# Enhancing Reasoning Capabilities by Instruction Learning and Chain-of-Thoughts for Implicit Discourse Relation Recognition

**Yuxiang Lu, Yu Hong*, Zhipang Wang, Guodong Zhou**

School of Computer Science and Technology, Soochow University, SuZhou, China

{destinylu, tianxianer, zhipangwang}@gmail.com, gdzhou@suda.edu.com

## Abstract

The aim of implicit discourse relation recognition is to comprehend the sense of connection between two arguments. In this work, we present a classification method that is solely based on generative models. Our proposed approach employs a combination of instruction templates and in-context learning to refine the generative model for effectively addressing the implicit discourse relation recognition task. Furthermore, we utilize Chain-of-Thoughts to partition the inference process into a sequence of three successive stages. This strategy enables us to fully utilize the autoregressive generative model's potential for knowledge acquisition and inference, ultimately leading to enhanced performance on this natural language understanding task. The results of our experiments, evaluated on benchmark datasets PDTB 2.0, PDTB 3.0, and the CoNLL16 shared task, demonstrate superior performance compared to previous state-of-the-art models.

## 1 Introduction

Discourse relation recognition refers to identifying the sense of the relation between two arguments. This task is categorized into two types: explicit discourse relation recognition (EDRR) and implicit discourse relation recognition (IDRR) depending on whether explicit connectives, such as "because" and "but", are present or absent between the argument pair. Our work investigates the potential of generative models and natural language generation for improving the performance of IDRR.

Recognizing implicit discourse relations involves comprehending and examining the semantic connections between argument pairs. Previous works have commonly employed semantic encoding to enhance the model's classification accuracy (Liu et al., 2020; Dou et al., 2021; Xiang et al., 2022a). Generative models have also been utilized

---

*Corresponding author

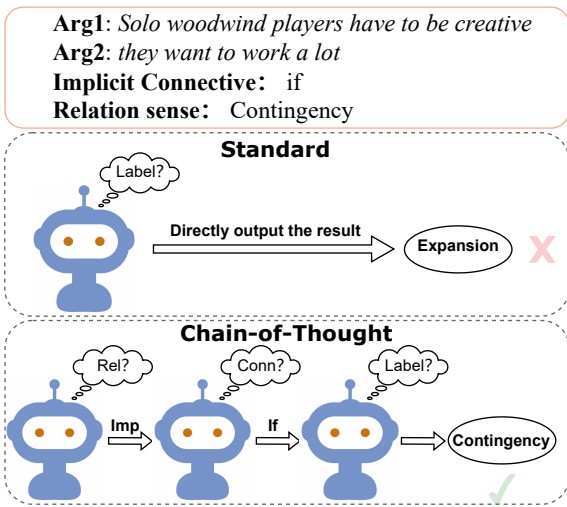

| |
| --- |
| **Arg1**: *Solo woodwind players have to be creative* |
| **Arg2**: *they want to work a lot* |
| **Implicit Connective**： if |
| **Relation sense**： Contingency |

Figure 1: Chain-of-Thoughts.

for IDRR. As an example, generation tasks have been used as an auxiliary task (Jiang et al., 2021) and in a limited form that restricts prompt learning (Zhou et al., 2022; Xiang et al., 2022b). With the emergence of large language models, there is a growing interest in utilizing generative models rather than encoder-only models for NLP applications. However, some studies also suggest that generic generative models do not perform as well as fine-tuning relatively small encoder-only models for NLU tasks (Qin et al., 2023). Our experiments also reveal that employing generative models to directly generate relation sense in the context of IDRR is an ineffective approach.

This work investigates how simple yet effective methods (IICOT) can unleash the inference capabilities of generative models. Figure 1 illustrates our approach of utilizing a thinking process to guide the model's output (COT). Specifically, we do not allow the model to output the relation sense directly; we compel the model to first identify whether the argument pair pertains to implicit or explicit relation data. This approach reduces the unwanted noise generated by explicit data. Next, the model

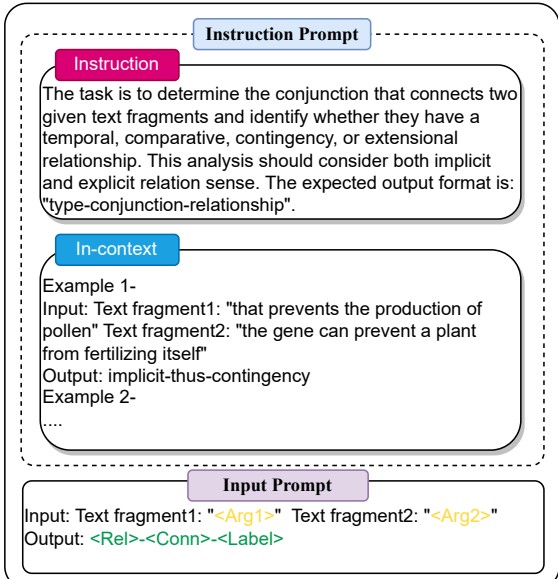

Figure 2: The full prompt template of our model.

identifies a reasonable conjunction between the argument pairs and bases its final inference on this analysis. To optimize the model's performance, we formulate a prefix prompt for better guidance which is in the form of instructions (I). By fine-tuning the instructions, we enhance the model's ability to learn and understand the task definition. Additionally, we employ the In-context learning (Min et al., 2022) (I) approach to provide additional examples to aid the model's comprehension of the prompt.

**Contributions:** Our work makes the following contributions. (a) We use generative methods on the IDRR task and explore methods for improving the inference power of generative models. (b) We investigate the impact of instruction learning, in-context learning, and Chain-of-Thoughts (COT) on the performance of generative models. Through our exploration, we are able to identify the causes and effects of these learning methods. (c) We achieve state-of-the-art performance on all three datasets, indicating the effectiveness of our approach.

## 2 Apporach

### 2.1 Instruction Learning

The primary objective of instruction fine-tuning is to enhance the language models' capacity to respond to natural language instructions. The method entails utilizing supervised signals to instruct language models on performing tasks described in instructions. As a result of instruction fine-tuning, language models learn to follow instructions and

respond to the same tasks. To test this approach, we devise instruction fine-tuning templates, as illustrated in Figure 2. The templates provide a comprehensive task definition for the model, enabling a deeper understanding of the task at hand. They utilize natural language to guide the model's thought process and restrict the format of the model's output to facilitate subsequent evaluation.

### 2.2 In-context Learning

In-context learning enables a language model to grasp a task and produce answers to queries based on given illustrative examples. Essentially, it entails training a proficient language model for estimating a conditional probability distribution model, relative to a specific condition. However, we have discovered in our research that providing the model with a specific number of instances during training enhances its adherence to format and facilitates more effective convergence. During our experiments, we meticulously prepare an example to represent each of the four relation senses, which is visually illustrated in Appendix A.

### 2.3 Chain-of-Thoughts

While regular training methods require models to tackle complex problems in a single step, people prefer an incremental approach, breaking down problems into smaller components to facilitate complex reasoning. This inclination towards incremental thinking enables people to engage in more nuanced and effective problem-solving. Our approach presents a simple yet effective method of prompting that mimics thinking process in the form of natural language prompts, as shown in Figure 2. Rather than providing a categorical answer directly, the model first considers whether the implied relationship is explicit or implicit. It then identifies appropriate connectives between pairs of arguments before finally providing the answer.

In the autoregressive generative mode, each token output by the model is influenced by its predecessors, creating a natural progression of thought. The reasoning process under standard generation prompt and with COT[1] is depicted in Figure 1. The latter method requires the model to provide a COT before producing a response. By incorporating COT into the prompting strategy, the performance

---

[1]The training set of PDTB 2.0 and PDTB 3.0 lacks explicit data on implicit chapter relations. To enhance the model's ability to generate the intended COT, a limited amount of explicit data was artificially incorporated.

| Model | PDTB 2.0 | | PDTB 3.0 | | CoNLL16 | |
|---|---|---|---|---|---|---|
| | Macro-F1 | Acc. | Macro-F1 | Acc. | Test Acc. | Blind Acc. |
| Baseline (roberta-base) | 57.65 | 65.50 | 63.32 | 65.50 | 61.11 | 62.42 |
| Baseline (roberta-large) | 64.03 | 70.46 | 67.52 | 71.44 | 64.52 | 68.13 |
| BMGF (Liu et al., 2020) | 63.39 | 69.06 | - | - | 65.15 | 72.21 |
| CVAE (Dou et al., 2021) | 65.06 | 70.17 | - | - | - | - |
| MANF (Xiang et al., 2022a) | - | - | 56.63 | 64.04 | - | - |
| CPrompt (Xiang et al., 2022b) | - | - | 70.88 | 75.17 | - | - |
| PCP (roberta-base)(Zhou et al., 2022) | 64.95 | 70.84 | - | - | 68.98 | 71.31 |
| PCP (roberta-large) (Zhou et al., 2022) | 67.79 | 73.80 | - | - | 72.36 | 74.51 |
| IICOT (flan-t5-base) | 65.26 | 71.13 | 69.79 | 73.98 | 69.84 | 72.38 |
| IICOT (flan-t5-large) | **69.23** | **76.04** | **73.06** | **77.46** | **73.46** | **75.84** |

Table 1: Main experiments on the 4-class classification in Macro-F1 and accuracy. The highest reported results of previous works are denoted by underlines.

| Model (PDTB 2.0) | Tem | Com | Con | Exp |
|---|---|---|---|---|
| BMGF (Liu et al., 2020) | 50.26 | 59.44 | 60.98 | 77.66 |
| JMCG (He et al., 2020) | 41.54 | 55.40 | 57.04 | 74.76 |
| CVAE (Dou et al., 2021) | 44.01 | 55.72 | 63.39 | 80.34 |
| PCP (Zhou et al., 2022) | 56.41 | 70.38 | 64.18 | 80.17 |
| IICOT (Ours) | **56.99** | **71.23** | **68.80** | **81.24** |
| Model (PDTB 3.0) | Tem | Com | Con | Exp |
| MANF (Xiang et al., 2022a) | 42.13 | 35.83 | 63.55 | 70.00 |
| IICOT (Ours) | **67.63** | **70.65** | **79.78** | **79.71** |

Table 2: The binary classification performance on PDTB 2.0 and PDTB 3.0 benchmark datasets.

| Model | Tem | Com | Con | Exp | M-F1 |
|---|---|---|---|---|---|
| Fine-tuning | 57.52 | 61.24 | 75.98 | 78.31 | 67.90 |
| +Instruction | 58.85 | 63.65 | 77.23 | 78.17 | 69.58 |
| +ICL | 58.50 | 62.93 | 77.83 | 77.41 | 69.23 |
| +ICL&Rel | 59.36 | 64.28 | 78.19 | 78.03 | 69.54 |
| +ICL&Conn | 59.66 | 69.33 | 78.30 | 78.62 | 72.97 |
| +Rel&Conn | 63.64 | 67.66 | 78.15 | 78.79 | 72.03 |
| IICOT (Ours) | **67.63** | **70.65** | **79.78** | **79.71** | **73.06** |

Table 3: We conduct ablation experiments on the latest PDTB3.0 dataset of IDRR, evaluating the F1 and Macro-F1 metrics in the binary and 4-way scenario.

of the model improves. All the specific prompt templates we used are in the Appendix B

## 3 Experiment

### 3.1 Experiment Settings

This study involves conducting experiments on three benchmark datasets: PDTB 2.0 (Prasad et al., 2008), PDTB 3.0 (Prasad et al., 2019), and CoNLL16 (Xue et al., 2016). Notably, the CoNLL16 dataset lacks manually annotated ligatures in 450 training data instances. To address this, we utilize the gpt-3.5-turbo model to predict the ligatures and establish them as the ground trut. Detailed statistics for each dataset and the settings of hyperparameters can be found in Appendix C. To ensure reproducibility, we will make all of our source code publicly availableh[2].

### 3.2 Main results

Table 1 and 2 present the results of our 4-way and binary classification performance on PDTB 2.0, PDTB 3.0, and CoNLL16 benchmark datasets. We employ the flan-t5 pre-trained model, a t5-based

[2]https://github.com/Destiny-Lu/IICOT_IDRR.

model specifically fine-tuned for a broad range of natural language processing tasks to better suit instruction learning. Remarkably, our approach outperforms current state-of-the-art methods in all datasets. Specifically, we achieve an impressive 4.62% increase in the Contingency category on PDTB 2.0, while realizing significant improvements in all of the categories on PDTB 3.0.

### 3.3 Ablation Experiments

Table 3 presents the findings of our ablation experiments, where **Fine-tuning** denotes that the input is only the argument pair while the direct output relation sense; **Instruction** indicates that the task definition directs the model to output labels; **ICL** refers to In-context learning; **Conn** refers to predicting connected words; **Rel** denotes predicting explicit or implicit data.

Our ablation experiments lead to four key conclusions: **First**, using instruction and ICL improves the model's performance in comparison to directly outputting results. Providing a certain amount of example enhances the model's understanding of the task. **Second**, appending Rel or Conn to ICL

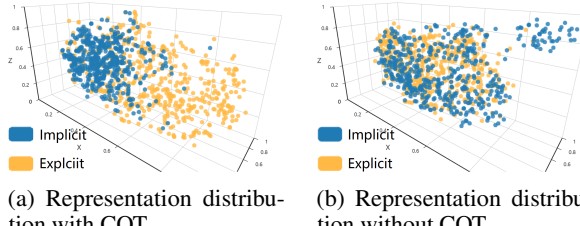

(a) Representation distribution with COT

(b) Representation distribution without COT

Figure 3: The noise reduction effect of COT is demonstrated through the distribution.

to predict chains before labels further improve the model's reasoning ability. **Third**, without ICL, adding both Rel and Conn also lead to a performance improvement. This indicates that the COT approach is highly effective. **Fourth**, the optimal value is reached by combining the approaches mentioned above, thereby demonstrating their individual validity and mutual reinforcement.

## 4 Analysis

### 4.1 Explicit Data

Our experiment of the CoNLL16 task reveals that incorporating both Explicit and Implicit judgments into the COT further enhances model performance. Accordingly, we propose the inclusion of explicit data in the PDTB dataset to scrutinize whether the performance improves. Our experimental findings (Appendix D) demonstrate that the performance improves upon the inclusion of a limited amount of explicit data. However, with increasing amounts of explicit data, the performance deteriorates substantially. This is because the distribution of explicit data differs from implicit data and the introduction of more explicit data results in amplified noise.

Figure 4 and 5 demonstrate that an increase in explicit data corresponds to a decrease in the model's accuracy in distinguishing between explicit and implicit data. Nonetheless, the model maintains a high accuracy rate of 96.83% and exhibits optimal performance at the 20% threshold of explicit data. These results suggest that the model remains relatively unaffected by noise and that successful data augmentation has been achieved.

### 4.2 How Chain-of-Thoughts works

**Denoising** It is believed that the model achieves denoising ability while generating COT. In order to demonstrate the denoising process, we analyze the output vectors of the model in terms of their representations. This allows us to gain insight into the denoising mechanism of the model.

The t-SNE method is frequently preferred for visualizing high-dimensional data, as it effectively presents local relationships and clustering structures. Specifically, t-SNE is adept at capturing similarities and differences within the data. Figure 3-(b) illustrates the model that incorporates **Rel** within the COT framework. This contrasts the COT model shown in Figure 3-(a), which does not feature these judgments. The generative model, such as flan-T5, lacks a dedicated token [CLS], unlike the BERT model. Consequently, in our study, we employ the encoding vector of the [BOS] token to represent the sentence after applying t-SNE dimensionality reduction. This approach enables us to examine the distribution pattern. The COT judgment effectively reduces noise and facilitates the acquisition of semantic knowledge pertaining to explicit data. The results demonstrate that the different types of data are well-separated upon the inclusion of COT judgment.

**Mitigating overfitting** It is our contention that the efficacy of COT also lies in its ability to alleviate model overfitting. Figure 6 presents data on training loss, loss on the development (dev) set during training, and changes in dev set performance for models trained both with and without COT. The table illustrates that abstaining from COT leads to a further drop in training loss, but also to a shift in dev set loss from low to high, along with a trend of decreasing performance. These phenomena suggest that the model overfits the training data.

COT enhances the informational content of model outputs. Although it increases the complexity of the task, this additional information effectively guides the model towards the desired direction, thereby mitigating the potential of overfitting.

## 5 Conclusion

We aim to enhance the reasoning capabilities of generative models in IDRR by employing a generation task framework and incorporating Instruction learning, in-context learning, and COT. Through our approach, we achieve a notable improvement over the baseline model, leading to state-of-the-art performance on three benchmark datasets.

In our future research, we plan to further investigate the utilization of generative models and even large language models. Specifically, we aim to explore the efficacy of larger models, including the

implementation of knowledge distillation to transfer knowledge from large models to smaller ones.

## Acknowledgment

The research is supported by National Key R&D Program of China (2020YFB1313601) and National Science Foundation of China (62376182, 62076174).

## 6 Limitations

The current design of our COT is not yet perfect, leaving ample room for improvement. Furthermore, our experimentation has been limited to the English corpus, without exploring other languages.

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

| | PDTB2.0 | | | | PDTB3.0 | | | | CoNLL16 | | | |
|---|---|---|---|---|---|---|---|---|---|---|---|---|
| **Implicit** | Tem. | Com. | Con. | Exp. | Tem. | Com. | Con. | Exp. | Tem. | Com. | Con. | Exp. |
| **Train** | 704 | 2,104 | 3,622 | 7,394 | 1,515 | 2,044 | 6,481 | 8,660 | 697 | 1,952 | 3,435 | 7,072 |
| **Dev./Blind** | 54 | 191 | 287 | 651 | 136 | 190 | 579 | 748 | 30 | 185 | 274 | 705 |
| **Test** | 68 | 146 | 276 | 556 | 148 | 154 | 529 | 643 | 36 | 88 | 122 | 276 |
| | PDTB2.0 | | | | PDTB3.0 | | | | CoNLL16 | | | |
| **Explicit** | Tem. | Com. | Con. | Exp. | Tem. | Com. | Con. | Exp. | Tem. | Com. | Con. | Exp. |
| **Train** | 2,904 | 4,674 | 2,792 | 5,342 | 3,435 | 5,104 | 3,356 | 8,634 | 2,752 | 4,383 | 2,579 | 5,008 |
| **Dev./Blind** | 248 | 431 | 277 | 506 | 348 | 422 | 249 | 748 | 318 | 404 | 254 | 503 |
| **Test** | 288 | 366 | 181 | 450 | 280 | 488 | 340 | 836 | 169 | 192 | 107 | 212 |

Table 4: Descriptive statistics of implicit discourse relation instances are reported for the datasets.

## A In-context Learning Examples

**Example 1-**
Argument1: there's a satisfaction in going against the rules
Argument2: he means the rule that a player can't cut it after a certain age
Relation sense: Extension

**Example 2-**
Argument1: that prevents the production of pollen
Argument2: the gene can prevent a plant from fertilizing itself
relation sense: Comparison

**Example 3-**
Argument1: he was heralded by a trumpet fanfare
Argument2: the judge marched down the center aisle in his flowing black robe
Relarion sense: Temporal

**Example 4-**
Argument1: however, the maximum coupon at which the notes can be reset is 16 1/4%
Argument2: the minimum coupon is 13 3/4%
Relarion sense: Comparison

## B Chain-of-Thoughts prompt

**Only label:**
Instruction: The task is to determine whether they have a temporal, comparative, contingency, or extensional relationship. This analysis should consider both implicit and explicit relationships.

**Conn & Label:**
Instruction: The task is to determine the conjunction that connects two given text fragments and identify whether they have a temporal, comparative, contingency, or extensional relationship. This analysis should consider both implicit and explicit relation sense. The expected output format is: "conjunction-relationship".

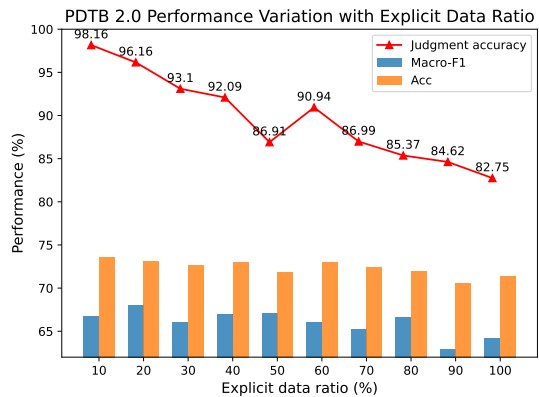

Figure 4: Experiments with adding explicit data on PDTB 2.0.

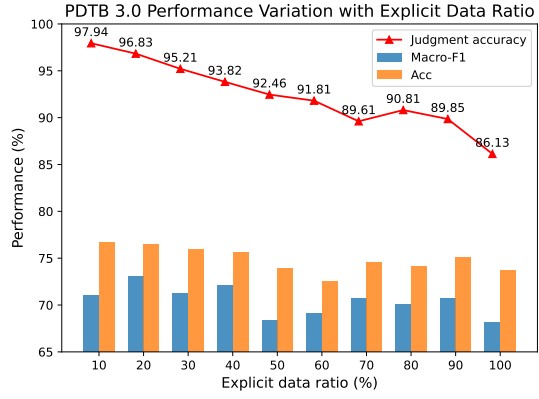

Figure 5: Experiments with adding explicit data on PDTB 3.0.

**Rel & Conn & Label:**
Instruction: The task is to determine the conjunction that connects two given text fragments and identify whether they have a temporal, comparative, contingency, or extensional relationship. This analysis should consider both implicit and explicit relation sense. The expected output format is: "type-

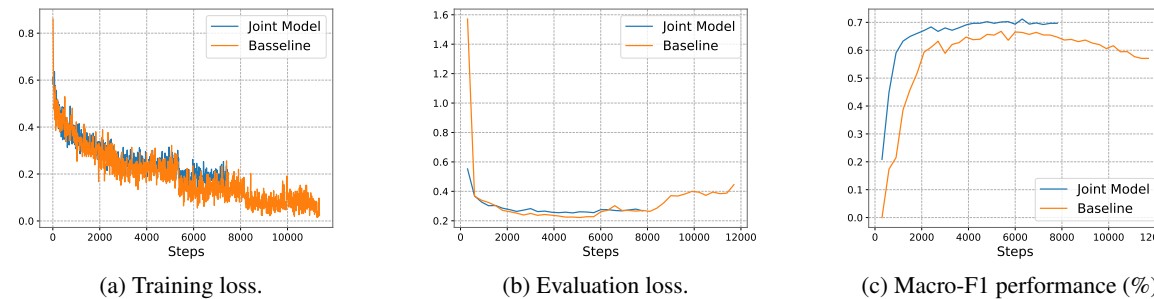

| (a) Training loss. | (b) Evaluation loss. | (c) Macro-F1 performance (%) |

Figure 6: The experimental records encompass training loss, evaluation set loss, and performance metrics.

conjunction-relationship".

## C  Dataset statistics and the hyperparameters

Table 4 presents the statistical information for specific datasets. In our evaluation process, we exclusively employ implicit data from the PDTB 2.0 dataset and PDTB 3.0 dataset. Conversely, for the CoNLL16 dataset, we utilize all available data in accordance with the official partitioning strategy.

During our training process, we employ a batch size of 16 and set the learning rate to 5e-5. We train the AdamW optimizer for 5 epochs, utilizing the default parameter settings. Moreover, we incorporate both warmup and linear learning rate decay strategies, with a warmup ratio of 0.1.

## D  Add Explicit Data

To examine the impact of explicit data on model performance, we gradually augment the amount of explicit data in the training sets of PDTB 2.0 and PDTB 3.0. The experiment results are detailed in Figure 4 and 5.

## E  Observation of Overfitting

The presence of overfitting can be readily observed by analyzing the training loss, evaluation set loss, and performance variation in the absence of the COT model. In contrast, the inclusion of the COT model leads to improved performance. It is important to highlight that the training steps differ between the two experiments due to the utilization of the early stop strategy.