# OpenReview forum: "Enhancing Reasoning Capabilities by Instruction Learning and Chain-of-Thoughts for Implicit Discourse Relation Recognition"
_EMNLP/2023/Conference — EMNLP 2023 Findings_

### Official Review · Reviewer_YJWg · 2023-07-19

**Soundness:** 3

**Excitement:**

4: Strong: This paper deepens the understanding of some phenomenon or lowers the barriers to an existing research direction.

**Paper Topic And Main Contributions:**

The paper titled "Enhancing Reasoning Capabilities by Instruction Learning and Chain-of-Thoughts for Implicit Discourse Relation Recognition" proposes a new method to approach the Implicit Discourse Relation Recognition (IDRR) task. The authors combine the use of generative models with a chain-of-thought model to determine the existing relation between arguments in the discourse. They evaluate their models considering three benchmarks: PDTB 2.0, PTDB 3.0, and CoNLL16.

**Questions For The Authors:**

Which evaluation metric is depicted in Table 2?

**Reasons To Accept:**

The proposed method combines generative models and prompting with an approach based on the chain-of-thought which looks original and interesting. Furthermore, the reported results outperform previous work considering both, Transformer-based approaches, prompting, and original methods.

**Reasons To Reject:**

The structure and the writing of some parts of the paper can be confusing. The fine-tuning / instruction learning section is not clear enough. Is the Instruction prompt used for fine-tuning the model in multiple epochs/iterations? Or is it provided at de beginning of every prompt without specific parameter fine-tuning? According to Appendix C there is a specific parameter fine-tuning, but this step is not clearly described in the paper (e.g., how is the fine-tuning task modelled. input, output, etc.).

Regarding the structure, it is a bit confusing that Figures 4 and 5 are analysed in the main body of the paper, but included in the appendix. Furthermore, they are mentioned before Figure 3 in the text. Figure 3 is also slightly confusing since (a) says with COT and (b) without COT, but (a) visualisation is definitely more noisy than (b), and then in the text is stated that:

"Figure 3-(b) illustrates the model that incorporates Rel within the COT framework. This contrasts the COT model shown in Figure 3-(a), which does not feature these judgments."

Which seems a bit contradictory with the caption of the figure.

Finally, a concerning aspect of this work is that the authors rely on ChatGPT to annotate some missing relations in the CoNLL16 benchmark without any further human validation: "Notably, the 148 CoNLL16 dataset lacks manually annotated ligatures in 450 training data instances. To address this, 150 we utilize the gpt-3.5-turbo model to predict the 151 ligatures and establish them as the ground truth."

**Reproducibility:**

2: Would be hard pressed to reproduce the results. The contribution depends on data that are simply not available outside the author's institution or consortium; not enough details are provided.

**Reviewer Confidence:**

4: Quite sure. I tried to check the important points carefully. It's unlikely, though conceivable, that I missed something that should affect my ratings.

---

> ### Author Rebuttal · Authors · 2023-08-29
>
> We are grateful for your thoughtful comments and valuable suggestions.
>
> **Com1:** The structure and the writing of some parts of the paper can be confusing. The fine-tuning / instruction learning section is not clear enough. Is the Instruction prompt used for fine-tuning the model in multiple epochs/iterations? Or is it provided at de beginning of every prompt without specific parameter fine-tuning? According to Appendix C there is a specific parameter fine-tuning, but this step is not clearly described in the paper (e.g., how is the fine-tuning task modelled. input, output, etc.).
>
>
> **Answer:** We sincerely apologize for the lack of clarity regarding the fine-tuning process in our paper. In fact, we did perform full fine-tuning on the flan-t5-base/large model using the method mentioned in the paper. As you've observed, we provide all the hyperparameter configurations and some training details in the appendix. Furthermore, we will make all training code publicly available to facilitate reproducibility.
>
>
> **Com2:** Regarding the structure, it is a bit confusing that Figures 4 and 5 are analysed in the main body of the paper, but included in the appendix. Furthermore, they are mentioned before Figure 3 in the text. Figure 3 is also slightly confusing since (a) says with COT and (b) without COT, but (a) visualisation is definitely more noisy than (b), and then in the text is stated that: "Figure 3-(b) illustrates the model that incorporates Rel within the COT framework. This contrasts the COT model shown in Figure 3-(a), which does not feature these judgments." Which seems a bit contradictory with the caption of the figure.
>
>
> **Answer:** Due to the page limit, we were unable to include Figure 4 and Figure 5 in the main body of the paper. We feel sorry to bring trouble on reading. In the revised version of this paper, we will rectify this problem by incorporating the necessary figures into the main body and accompanying the discussions with them.
>
>
> Furthermore, I sincerely apologize for the error in the order of Figures 3-(a) and 3-(b). We will correct this mistake in the extended version.
>
>
> **Com3:** Finally, a concerning aspect of this work is that the authors rely on ChatGPT to annotate some missing relations in the CoNLL16 benchmark without any further human validation: "Notably, the 148 CoNLL16 dataset lacks manually annotated ligatures in 450 training data instances. To address this, 150 we utilize the gpt-3.5-turbo model to predict the 151 ligatures and establish them as the ground truth."
>
>
> **Answer:** We fully understand your concern about the use of ChatGPT and human validation. We implemented strict annotation guidelines to ensure ChatGPT's responses were contextually appropriate. Specifically, we provided ChatGPT with argument pairs and relation categories. On this basis, we offered a list of 10 most commonly-used connectives per relation type for prompting ChatGPT, further constraining the responses. Following this process, we manually reviewed around 20 generated results for each relation category. Based on our observations, we deemed ChatGPT's generated connectives to be contextually appropriate.
>
> We appreciate your question and, accordingly, will conduct a comprehensive human evaluation of these 450 training instances, and provide a direct insight into the results generated by ChatGPT.
>
>
> **Q1:** Which evaluation metric is depicted in Table 2?
>
>
> **Answer:** We sincerely apologize for the lack of clear explanation regarding the evaluation metric for Table 2 in our paper. In our paper, the performance for all binary classification tasks is measured using F1-score.

---

### Official Review · Reviewer_XvkT · 2023-08-03

**Soundness:** 3

**Excitement:**

3: Ambivalent: It has merits (e.g., it reports state-of-the-art results, the idea is nice), but there are key weaknesses (e.g., it describes incremental work), and it can significantly benefit from another round of revision. However, I won't object to accepting it if my co-reviewers champion it.

**Paper Topic And Main Contributions:**

This work explores recent generative AI models on the implicit discourse parsing task.

**Questions For The Authors:**

Suggestions

* More empirical datapoints

1) Comparison between different generative AI models such as Llama and Flan-T5
2) Comparison between different sizes of models such as 3B, 7B, and 11B

* More analysis of evaluations

What does it make still recent language models suffer difficulties in this task? (low performance boosting?)

**Reasons To Accept:**

It is an interesting empirical work to guide other studies further.

**Reasons To Reject:**

* Lack of enough empirical datapoints.

Since it is a purely empirical work employing common techniques to this task, the value of this work relies on enough empirical findings and datapoints

* Lack of analysis for evaluations

It is interesting to see the performance boosting is not remarkable such as 1% accuracy improvement. What makes this low boosting? Is it because of the task setup or these recent language models are not enough to deal with it?

**Reproducibility:**

3: Could reproduce the results with some difficulty. The settings of parameters are underspecified or subjectively determined; the training/evaluation data are not widely available.

**Reviewer Confidence:**

3: Pretty sure, but there's a chance I missed something. Although I have a good feel for this area in general, I did not carefully check the paper's details, e.g., the math, experimental design, or novelty.

---

> ### Author Rebuttal · Authors · 2023-08-29
>
> We are grateful for your insightful comments and thoughtful suggestions.
>
>
> **Com1:** Since it is a purely empirical work employing common techniques to this task, the value of this work relies on enough empirical findings and datapoints
>
>
> **Answer:** We realize your concern and regard it as the use of scenario learning and instruction learning. Indeed, they are commonly-used techniques recently. However, we take a keen interest in exploring possibility of leveraging generative models for IDRR, where scenario learning and instruction learning are just supportive techniques. To our best knowledge, there hasn’t yet any previous work dealing with IDRR within a generative framework.
>
> In addition, the other difference of our approach from the previous work is to bring a new COT into the denoising process. The design of COT is truly grounded on the investigation of data (positive cases and noises), though it is not tailored but on the contrary useful for dealing with similar problems. In the area of utilizing generative LLMs, the design of COT has been a hot topic, because a well-crafted COT can significantly boost the reasoning abilities of LLMs. Our study proposes a simple yet effective COT.
>
>
> **Com2:** It is interesting to see the performance boosting is not remarkable such as 1% accuracy improvement. What makes this low boosting? Is it because of the task setup or these recent language models are not enough to deal with it?
>
>
> **Answer:** It's true that the performance improvement on CoNLL is not pronounced. This attributes to two aspects as below. First, the performance of the baseline model on CoNLL has been at a considerably high level. Second, the generative models like flan-t5-large (with around 800M parameters) are not enough large to treat with the complicated problems in IDRR. In fact, Chunkit Chan's paper [1] provides comprehensive testing of ChatGPT on this task, and they prove that ChatGPT performs quite modestly. In light of this, as you suggested, we will explore larger models in our future work.
>
> **Q1:** More empirical datapoints. Comparison between different generative AI models such as Llama and Flan-T5. Comparison between different sizes of models such as 3B, 7B, and 11B
>
>
> **Answer:** We plan to conduct experiments with larger models such as llama-7b and llama-13b in the future work. However, for a fair comparison with previous work, we are currently validating our approach on the smaller models.
>
>
> **Q2:** More analysis of evaluations
>
>
> **Answer:** We appreciate the suggestion for more in-depth analysis of our evaluations. In our reviewed paper, we plan to delve deeper into the comprehensive performance analysis. This will include investigating specific error cases, identifying patterns in misclassifications, and exploring the impact of various factors on the model's predictions. By conducting a more extensive analysis, we aim to offer valuable insights into the strengths and limitations of our approach, which will contribute to a richer discussion of our results.
>
>
> **Q3:** What does it make still recent language models suffer difficulties in this task? (low performance boosting?)
>
>
> **Answer:** Good question. IDRR is that a challenging task which requires a machine to deeply understand semantics of arguments, and meanwhile sufficiently learn much richer knowledge about discourse structures. In some cases, we nearly observed that some hypothesis needs to be considered first for human to correctly determine the relations, without giving reliable clues in the texts. More importantly, the arguments are long and edited by different persons who have inconsistent writing habits and distinct desires. As a result, any subtle changes made on arguments allow most of the relation types to be explainable, from the perspectives of different readers. By contrast, the existing LLMs are more capable of generating readable texts using fluent languages, though they haven’t yet been proven to possess all-sided capacities as mentioned above, while such capacities are required in IDRR. Transfer learning and knowledge distillation may enhance LLMs, and allow them to work as the tailored versions. However, the available data is so sparse that the comprehensive capacity of IDRR cannot be developed using the data.

---

### Official Review · Reviewer_znNK · 2023-08-03

**Typos Grammar Style And Presentation Improvements:** The writing should be improved.
**Soundness:** 3

**Excitement:**

3: Ambivalent: It has merits (e.g., it reports state-of-the-art results, the idea is nice), but there are key weaknesses (e.g., it describes incremental work), and it can significantly benefit from another round of revision. However, I won't object to accepting it if my co-reviewers champion it.

**Missing References:**

- Wangqiu Long, Bonnie Webber. Facilitating Contrastive Learning of Discourse Relational Senses by Exploiting the Hierarchy of Sense Relations. In EMNLP 2022.

**Paper Topic And Main Contributions:**

This paper explores a generative model for the IDRR task inspired by recent work about Instruction learning, In-context Learning, and Chain-of-thought. The authors provide extensive experiments to show the effectiveness of the method. While the general idea is intriguing, there is room for improvement in both the statements and experiments.

**Questions For The Authors:**

Questions:
- 1. I don't understand how your work deals with the explicit case during training. Did you input only the arguments of an explicit example to the model? And then also predict the relation type, connective, and label?
- 2. I doubt your implementation of the RoBERTa baseline is correct. In papers [1][2], they reported a much better performance of the Roberta baseline than your paper. For example, the RoBERTa's performance in paper[2] for PDTB 3.0 level1 is 73.51/67.98 for Acc/F1, which is almost the same as your model's performance.
- 3. I doubt your statement in line 83. I find paper[3] performs better than your method, especially on the F1 score. I don't think sota is necessary for this work. But your statement about sota bothers me a lot.
- 4. In Figure 3, I find the representation without COT seems to be better separated than with COT. But your description in lines 240-242 makes me quite confused. Why do you think with COT is well-seperated?

References:
- [1] Yuxin Jiang, Linhan Zhang, and Wei Wang. Global and Local Hierarchy-aware Contrastive Framework for Implicit Discourse Relation Recognition. In ACL 2023 Findings.
- [2] Wei Liu and Michael Strube. Annotation-Inspired Implicit Discourse Relation Classification with Auxiliary Discourse Connective Generation. In ACL 2023.
- [3] Wangqiu Long, Bonnie Webber. Facilitating Contrastive Learning of Discourse Relational Senses by Exploiting the Hierarchy of Sense Relations. In EMNLP 2022.

**Reasons To Accept:**

- An interesting model for IDRR task.
- Experimental results show the effectiveness of the method.

**Reasons To Reject:**

- Some descriptions are not clear (refer to the question).
- Experimental results of the RoBERTa baseline seems incorrect.
- They claimed sota but actually not.

**Reproducibility:**

3: Could reproduce the results with some difficulty. The settings of parameters are underspecified or subjectively determined; the training/evaluation data are not widely available.

**Reviewer Confidence:**

4: Quite sure. I tried to check the important points carefully. It's unlikely, though conceivable, that I missed something that should affect my ratings.

---

> ### Author Rebuttal · Authors · 2023-08-29
>
> We are grateful for your time in reviewing our submission.
>
> We observe that part of the questions is consistent with the comments for rejection. Therefore, we directly respond to the questions.
>
>
> **Q1:** I don't understand how your work deals with the explicit case during training. Did you input only the arguments of an explicit example to the model? And then also predict the relation type, connective, and label?
>
>
> **Answer:** Absolutely. For the explicit data, we merely input the explicitly-related arguments after removing the inner explicit connectives. Therefore, our approach is able to predict the triple of type, connective and label for COT.
>
>
> **Q2:** I doubt your implementation of the RoBERTa baseline is correct. In papers [1][2], they reported a much better performance of the Roberta baseline than your paper. For example, the RoBERTa's performance in paper[2] for PDTB 3.0 level1 is 73.51/67.98 for Acc/F1, which is almost the same as your model's performance.
>
>
> **Answer:** True. In some of the previous work, the substantially better performance of the RoBERTa baseline is reported. We feel sorry to overlook the studies in [1] and [2]. Though, we find that they are published after the deadline of EMNLP 2023. We are willing to include the citations of them in the revised paper.
>
> There are a variety of factors causing significantly inconsistent performance of the RoBERTa baseline, such as the training environment and hyperparameter configuration. In our case, we mainly carry out experiments using flan-T5. For environmental sustainability reasons, we did not extensively fine-tune the RoBERTa baseline. Therefore, it didn’t show an optimal performance.
>
> Our baseline model is implemented using the Hugging Face trainer, and we will make all training code publicly available for reproducibility and reference.
>
>
> **Q3:** I doubt your statement in line 83. I find paper[3] performs better than your method, especially on the F1 score. I don't think sota is necessary for this work. But your statement about sota bothers me a lot.
>
> **Answer:** Indeed, our performance is lower compared to Long et al's work [3]. However, we observe that Long et al employed data augmentation. This significantly boosted their classification performance for temporal relation labels by nearly 20% compared to SOTA. We believe that the existing approaches may benefit from this attempt, and therefore, take a keen interest in adding it to our study in the extended paper. We promise to pay more attention to compare our enhanced approach with Long et al’s work, and provide sufficient discussion on the achievements.
>
>
>
>
> **Q4:** In Figure 3, I find the representation without COT seems to be better separated than with COT. But your description in lines 240-242 makes me quite confused. Why do you think with COT is well-seperated?
>
>
> **Answer:** We apologize for the oversight. The left side of Figure 3 does not have COT, while the right side does.

---

### Meta-Review · Area_Chair_qaXd · 2023-09-16

**Recommendation:** 3

**Metareview:**

This paper examines the implicit discourse relation recognition (IDRR) task, proposing a generative model that exploits instruction learning, chain-of-thought, and in-context learning. Overall, the paper does not have serious flaws. While one reviewer found this work exciting, the remaining two reviewers were not particularly excited about it.

Strengths:

The use of instruction learning and chain-of-thought for the IDRR task is new and interesting.

Extensive experiments were conducted.

Weaknesses:

The reviewers agreed that some parts of the paper were confusing (e.g., instruction learning).

There was missing related work (e.g., the Long and Webber 2022 paper).

The empirical contribution of this paper could be strengthened. For instance, there wasn't any empirical analysis of the results (e.g., the seemingly small improvements). In addition, the claims could be strengthened through experiments with different models and different model sizes. The rebuttal did not provide additional empirical results or any analysis of the results.

The reviewers did not change their scores after reading the rebuttal and believed the paper could benefit from another round of revision.

---

### Decision · Program_Chairs · 2023-10-07

**Decision:**

Accept-Findings

**Comment:**

This paper examines the implicit discourse relation recognition (IDRR) task, proposing a generative model that exploits instruction learning, chain-of-thought, and in-context learning. Overall, the paper does not have serious flaws. While one reviewer found this work exciting, the remaining two reviewers were not particularly excited about it.

Strengths:

The use of instruction learning and chain-of-thought for the IDRR task is new and interesting.

Extensive experiments were conducted.

Weaknesses:

The reviewers agreed that some parts of the paper were confusing (e.g., instruction learning).

There was missing related work (e.g., the Long and Webber 2022 paper).

The empirical contribution of this paper could be strengthened. For instance, there wasn't any empirical analysis of the results (e.g., the seemingly small improvements). In addition, the claims could be strengthened through experiments with different models and different model sizes. The rebuttal did not provide additional empirical results or any analysis of the results.

The reviewers did not change their scores after reading the rebuttal and believed the paper could benefit from another round of revision.